# Ultrafast data mining of molecular assemblies in multiplexed high-density super-resolution images

Yandong Yin [1], Wei Ting Chelsea Lee [1] & Eli Rothenberg[1]

Multicolor single-molecule localization super-resolution microscopy has enabled visualization of ultrafine spatial organizations of molecular assemblies within cells. Despite many efforts, current approaches for distinguishing and quantifying such organizations remain limited, especially when these are contained within densely distributed super-resolution data. In theory, higher-order correlation such as the Triple-Correlation function is capable of obtaining the spatial configuration of individual molecular assemblies masked within seemingly discorded dense distributions. However, due to their enormous computational cost such analyses are impractical, even for high-end computers. Here, we developed a fast algorithm for Triple-Correlation analyses of high-content multiplexed super-resolution data. This algorithm computes the probability density of all geometric configurations formed by every triple-wise single-molecule localization from three different channels, circumventing impractical 4D Fourier Transforms of the entire megapixel image. This algorithm achieves $10^2$-folds enhancement in computational speed, allowing for high-throughput Triple-Correlation analyses and robust quantification of molecular complexes in multiplexed super-resolution microscopy.

[1] Department of Biochemistry and Molecular Pharmacology, New York University School of Medicine, New York, NY 10016, USA. Correspondence and requests for materials should be addressed to Y.Y. (email: Yandong.Yin@nyumc.org) or to E.R. (email: Eli.Rothenberg@nyumc.org)

Single Molecule Localization Microscopy (SMLM)[1–3] has emerged as a leading super-resolution imaging approach for nanoscale visualization of molecular structures in cells. SMLM achieves ~10 nm spatial resolution by stochastically sampling a small subset of fluorophores within a dense sample, and localizing each of these fluorophores with precise Point Spread Function (PSF) fitting. By encoding more single-molecule information into their PSFs, SMLM has been further extended with various advanced features[4–6]. Among these features, the multi-color SMLM[7,8] leveraged its potential for in situ probing of the geometric configuration of specific molecular assemblies by mapping two or more of their key components that are specifically labeled with different fluorophores[9,10].

Given the improved capabilities and specificity of molecular labeling schemes in addition to nanoscale spatial resolution, multi-color SMLM is poised to become the leading approach for visualizing specific molecular assemblies in cells. However, improved identification and analyses of specific molecular assemblies are required when imaging regions of interest within crowded cellular features. Recent developments of density-based Bayesian clustering[11], Ripley's K-function variants[12–14], as well as unsupervised clustering algorithms such as DBSCAN[15], OPTICS[16], and Delaunay Triangulation-based method[17], have provided unbiased and robust analyses of single-color SMLM images with high molecular density. However, approaches for quantifying the spatial correlations among different molecular species, especially in densely distributed molecules merged with background noises, remains particularly challenging.

The Pair- and Triple-correlation functions have been developed for analyzing organizations of molecular complexes in their dense distributions[18–20]. The correlation functions probe the probability densities of the presence of every pair- or triple-wise localization of molecules from the same (Auto-Pair-Correlation, Auto-PC) or different channels (Cross-Pair-Correlation, Cross-PC, and Triple-Correlation, TC), and examine the significance of such local probability density over its global stochastic fluctuations. It is thus robust in deciphering the spatial correlation from their stochastic and dense distributions. While the Cross-PC function evaluates the significance of identifying constant molecule-molecule interaction between two species, the TC function probes the geometric relationship among three molecular species, and conveys higher-level information with respect to the spatial organization of molecular assemblies[20]. As such the TC function provides a new class of analyses that is distinct from lower-order correlation analyses and cannot be obtained via combination of three pair-wise cross-correlation between three different molecules as A&B, A&C, and B&C, for the purpose of identifying the assembly of ABC complexes[14]. Currently, the most common method for computing the Triple-Correlation function is based on the bispectrum convolution theorem, which requires the detected localizations to be rendered from coordinates into binary pixels, and performs a 4D discrete Fourier Transform (FT)[21]. However, such 4D pixel-to-pixel FT lead to massive escalation in the computation cost by an order of $\sim(N \log_2 N)^4$ ($N$ is the SR image size along one dimension in the unit of rendered pixel), resulting in impractical computation requirements for even high-end computers. Importantly, these computational limitations cannot be overcome by rendering coordinates into bigger pixels nor by analyzing smaller subsets of the image, both of which would likely results in reduced statistical accuracy during Fourier Transform. Here we address these limitations by developing a direct coordinate-based Triple-Correlation algorithm (dTC). This algorithm preserves the theoretical accuracy of SMLM and minimizes the computational cost to a feasible level. The presented algorithm is validated by both simulation and experimentally and complied into Matlab executable functions for both CPU and GPU implementation.

## Results

**Formulation of the dTC algorithm.** The dTC algorithm is based on the binary nature of an SMLM image, spanning the vector-based data structure (list of molecular localizations) to perform correlation analyses across coordinates from the different channels, as illustrated in Fig. 1a–c (see Supplementary Note 1 for detailed derivations). The TC function is defined as:

$$g(\mathbf{r}_{12}, \mathbf{r}_{13}) = \frac{\langle \rho_{CH1}(\mathbf{R}) \rho_{CH2}(\mathbf{R} + \mathbf{r}_{12}) \rho_{CH3}(\mathbf{R} + \mathbf{r}_{13}) \rangle_{\mathbf{R}}}{\langle \rho_{CH1}(\mathbf{R}) \rangle_{\mathbf{R}} \langle \rho_{CH2}(\mathbf{R}) \rangle_{\mathbf{R}} \langle \rho_{CH3}(\mathbf{R}) \rangle_{\mathbf{R}}}, \quad (1)$$

where $\langle \rangle_R$ denotes averaging over all positions $\mathbf{R}$ in the image; $\rho(\mathbf{R}) = \left( \int_{\Delta S} f(\mathbf{R}) \right) / \Delta S$ is the local density at $\mathbf{R}$ within a differential area $\Delta S$. Considering the sparsity of the SMLM data (i.e., a set of localization coordinates $\mathbb{C}$, and thus $\rho(\mathbf{R}) = 0$, if $\mathbf{R} \notin \mathbb{C}$), the dTC computes $g(\mathbf{r}_{12}, \mathbf{r}_{13})$ directly, according to its definition (Eq. 1) (Supplementary Note 1). In brief, iterative analysis in this system is preformed such that each localization coordinate of a given channel (R-Red) serves as a vector origin; the local density of the coordinates from the other two channels (B-Blue, G-Green) at certain distances $\mathbf{r}_{RB} = (r_{RB}, \theta)$ and $\mathbf{r}_{RG} = (r_{RG}, \theta + \Delta\theta)$ are multiplied and the product is averaged over the entire angular space $\theta$ (Fig. 1a). Such computation is repeated and integrated as each Red localization is visited as an origin, followed by normalizing the result with the size of the canvas and the average density of the three molecular species (Fig. 1b). The Triple-Correlation profile is then transformed and represented as a function of a set three pair-wise distances $\{r_{RB}, r_{RG}, r_{BG}\}$ (Fig. 1c, Supplementary Note 1, and Supplementary Figure 2)[22]. To further interpret the triple-correlation results, we analyzed the local maxima of the calculated correlations which represent the most significant geometric configurations within the coordinate system of the three colors. The configurations are then displayed as triangles in which the size of the circle at the vertex denotes the correlation amplitude (Fig. 1d).

**Validation of the dTC algorithm.** To test the dTC algorithm, we simulated an SMLM image in which two types of molecular patterns formed by three different species were mixed and randomly positioned and oriented onto an ~10 × 10 μm$^2$ canvas (Fig. 2a). The simulated coordinates (~$10^4$ coordinates / channel, ~100 coordinates / μm$^2$ / channel) from three channels were then submitted to the dTC algorithm. The TC profile generated by the dTC algorithm displays two significant local maximums, representing the two different molecular configurations in the simulated image (Fig. 2b). We note that instead of computing the Triple-Correlation in a redundant pixel-by-pixel visiting manner as the typical Fourier Transform algorithm (ftTC), the dTC algorithm accomplishes the computation by directly visiting each coordinate in the reference channel. This features two advantages of dTC over the ftTC algorithm: 1) coordinates are no longer rendered into approximate pixels and thus dTC reaches the theoretical accuracy in calculation, and 2) the coordinate-visiting method for dTC takes much less time than the pixel-visiting method for ftTC. We simulated a series SMLM images with fixed canvas size but different molecular densities. Although the dTC algorithm consumed more computation time when the image became denser, it was in general much less time-consuming than the ftTC calculation (Fig. 2c), which is on the order of $\sim(N \log N)^4$ through bispectrum[20].

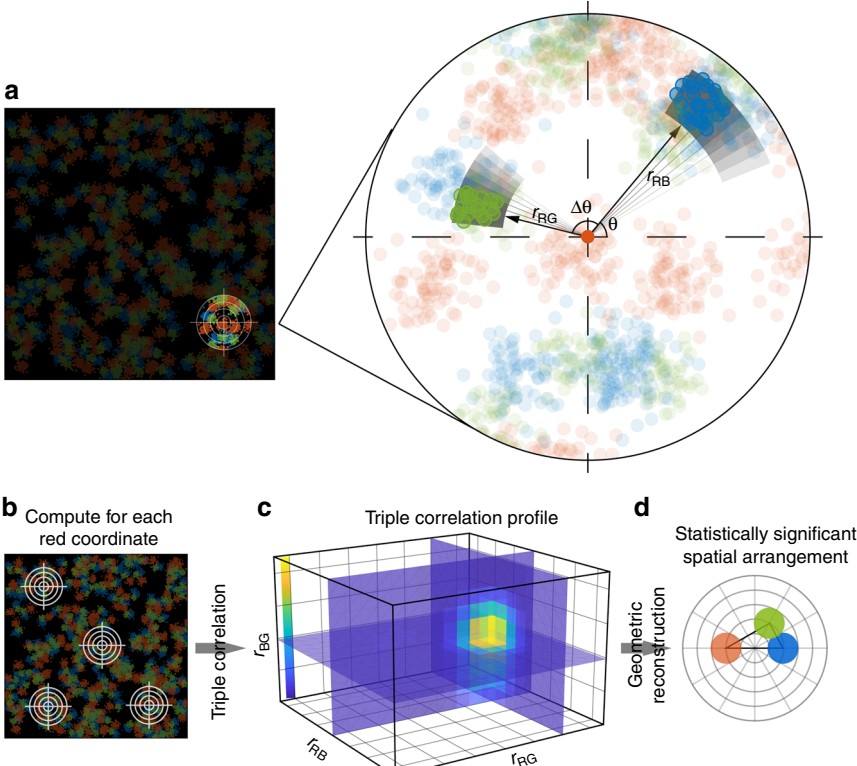

**Fig. 1** Schematic illustration of the direct TC algorithm. **a** Schematic illustration of dTC computation. Considering three sets of coordinates from blue, red, and green channel, the triple-correlation is calculated as: Each coordinate $\mathbf{r}_i^{CH_R}$ in the red channel is visited as origin $O$ (zoom-in, highlighted red). The density of blue coordinates positioning at $\mathbf{r}_{RB} = (r_{RB}, \theta)$ (highlighted blue) within the differential area ($\Delta S = r_{RB}\, d\theta dr$) and that of the green coordinates positioning at $\mathbf{r}_{RG} = (r_{RG}, \theta + \Delta\theta)$ (highlighted green) within the differential area ($\Delta S = r_{RG}\, d\theta dr$) are calculated and multiplied. The product is then integrated along $\theta$ through $[0, 2\pi]$ or appropriate range $[\varphi_1, \varphi_2]$ for edge correction (Supplementary Figure 1). **b** The same calculation in **a** is iteratively performed and integrated as each red coordinate is visited as an origin, followed by normalizing with the size of the canvas and the average molecular density of the three molecular species. **c** The computed triple-correlation profile is then plotted as a function of the triple-wise distances $\{r_{RB}, r_{RG}, r_{BG}\}$. **d** The local maximum(s) significantly higher ($\geq$ mean + 2.5 SD) than the fluctuation of the triple-correlation profile (**c**) is found at certain $\{r_{RB}^{max}, r_{RG}^{max}, r_{GB}^{max}\}$ and drawn as a triangle that represents probable geometric configuration among these three species. The size of the circles at the vertexes represent the correlation amplitude

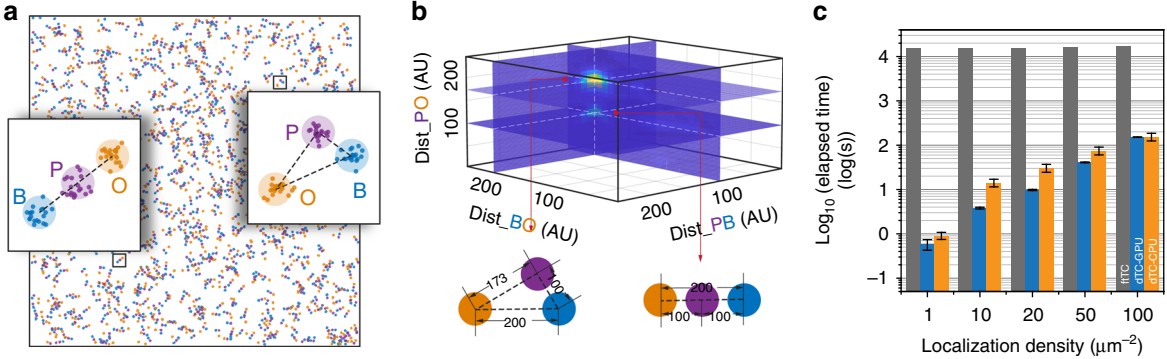

**Fig. 2** Validation of the dTC algorithm on simulated SMLM data. **a** Simulated three-color single-molecule localization image in which two different molecular patterns were mixed rand randomly orientated. ~ 1000 such patterns in total were seeded onto 10240 × 10240 nm$^2$ canvas, and each pattern has ~10 localizations in each channel. **b** The triple-correlation profile in which two local maxima were found representing the two input geometric configurations. **c** Comparison between the ftTC and dTC algorithm in computing the triple-correlation. Three-color images of the same canvas size (10,240 × 10,240 nm$^2$) but different molecular densities were submitted for dTC computation. These coordinates were rendered onto 2048 × 2048 pixelized image (pixel size = 5 × 5 nm$^2$) for ftTC computation. Error Bars in dTC computation stands for the SD of five computation replicates while ftTC computation was only performed once due to its computational costs. Source data for Fig. 2c are provided as a Source Data file

We next validated the dTC algorithm on experimental multi-color SMLM images of DNA replication fork associated proteins in nuclei of U2OS cells. The replisome is the fundamental unit of the replication machinery, within which the replicative helicase, MCM, and polymerases are the essential components that perform DNA replication. With MCM unwinding parental double-stranded DNA, the polymerases synthesize daughter strands along the unwound parental DNA template, while RPA

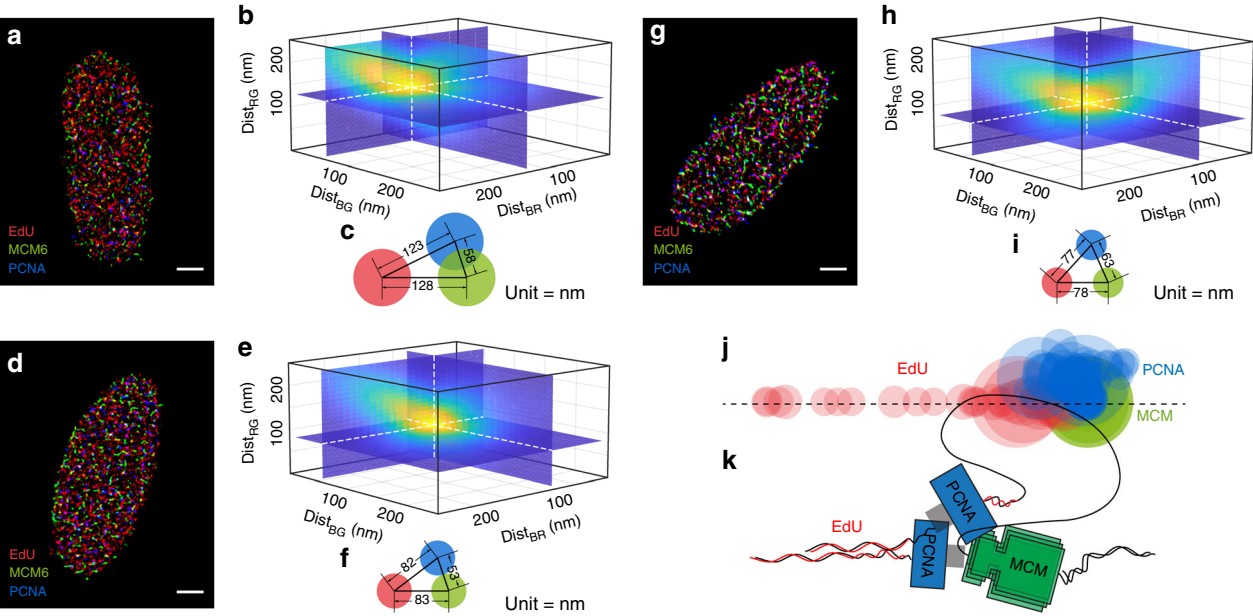

**Fig. 3** Validation of the dTC algorithm on experimental SMLM data. **a** A representative U2OS nucleus labeled with EdU (R), PCNA (B), and MCM (G). The intrinsic configuration amongst RGB is barely seen by eye due to the highly dense complexity. **b**, **c** Triple-Correlation (**b**) and its reconstructed EdU-PCNA-MCM configuration (**c**) of the nucleus in **a**. **d–i** Two other representative U2OS nuclei (**d**, **g**) and their triple-correlation profile. Scale bar = 2500 nm in **a**, **d**, **g**. **j** Configurations resolved from $N = 86$ nucleus in total from 3 experimental replicates are filtered by the criteria Dist_BG (distance between PCNA and MCM) ≤ 100 nm and overlaid by aligning the RG edge onto the same horizontal, so that the relative distribution of PCNA (B) to EdU (R) and MCM (G) is clearly seen to form an EdU-PCNA-MCM sequential pattern. The circle size of at each vertex displays the triple-correlation amplitude. **k** A molecular model of the sequential EdU-PCNA-MCM configuration. Source data for Fig. 3j are provided as a Source Data file

coats onto the transient single-stranded DNA (ssDNA) between polymerases and helicases[23]. Immunofluorescence techniques can be used to highlight the position of MCM and the polymerases involved in replication. EdU can also be incorporated into newly synthesized DNA and labeled with a fluorophore. In an individual mammalian replication fork, the helicase MCM, the polymerases-PCNA complex, and EdU display a sequential EdU-PCNA-MCM configuration and provide a unique molecular layout for validating the performance of triple-correlation (dTC) in dense distributions[10,20].

To examine the dTC algorithm with this EdU-PCNA-MCM configuration, we pulse labeled nascent DNA with EdU for 15 min before cell fixation, and then label PCNA and MCM using fluorophore-conjugated primary antibodies. Figure 3a shows three representative U2OS nuclei labeled with EdU (Red, R), PCNA (Blue, B), and MCM (Green, G). We note that the dense distribution of the three species makes it impractical to visually identify how these species are spatially configured within an individual replication unit. Figure 3b displays the nucleus' dTC as a function of the three pair-wise distances between the three molecular species; the intrinsic spatial configuration (Fig. 3c) was identified by localizing the different localization distances of the different channels where their dTC reaches its local maximum (indicating the most probable configurations). To generate a comprehensive triangulated map of the relative positioning among PCNA (B), EdU (R), and MCM (G), we analyzed 86 nuclei, and aligned the resolved EdU-PCNA-MCM configurations onto the same EdU-MCM horizon. Figure 3j shows the overlaid configurations, indicating that the sequential configuration EdU-PCNA-MCM is easily obtained via the triple-correlation function, even in dense images where these features are not readily observed (Fig. 3h). The dTC algorithm was also validated by examine the EdU-RPA-MCM molecular layout (Supplementary Figure 3).

**TC function estimates RPA density at each Replication Fork.** A significant advantage of Triple-Correlation is that it provides the average probability density of finding specific molecular species at a given molecular configuration. Given a seemingly amorphic complex ABC (composed of different units A, B, and C), the Triple-Correlation describes the average probability density of finding molecule-B at distance $r_{AB}$ away AND molecule-C at distance $\mathbf{r}_{BC}$ originated at each molecule A. By dividing the Triple-Correlation by the observed probability of finding B $r_{AB}$ away from each molecule A, one can obtain the posterior probability that C is found associated with a given sub-complexes AB at a certain configuration. To test this approach, we quantified the different replisome configurations arising from external perturbations due to Aphidicolin (APH) treatment, inducing abnormally lower polymerase activity, resulting in the global accumulation of Replication Protein A (RPA) at replication sites (termed replication foci) (Supplementary Figure 4). Such APH-induced recruitment of RPA is attributed to the initiation of new replication forks (replication origins), compensating for the APH inhibited forks[24], though it remain unclear whether (and how) APH affects the specific distribution of RPA at individual forks due to the limitations of current techniques as well as the data mining capabilities. Employing the dTC algorithm onto the multiplexed SMLM image of PCNA, MCM, and RPA (Fig. 4a–c) enables quantification of the local distribution of RPA at each replication fork. In brief, the SMLM coordinates of PCNA, MCM, and RPA were firstly submitted to the dTC algorithm to compute the Triple-Correlation profile and identify its local maxima as their intrinsic configuration at the given correlation distances. The conditional probability of finding RPA with each given PCNA-MCM fork $P$ (RPA|PM) was then calculated by dividing the Cross-Correlation between PCNA and MCM at the distance of $d_{PM}$ from $g_{TC}^{max}$. Finally, the local density at each PCNA-MCM fork $\rho$ (RPA|PM) was quantified by multiplying the overall average RPA density $\rho_{RPA}$ with the local conditional probability

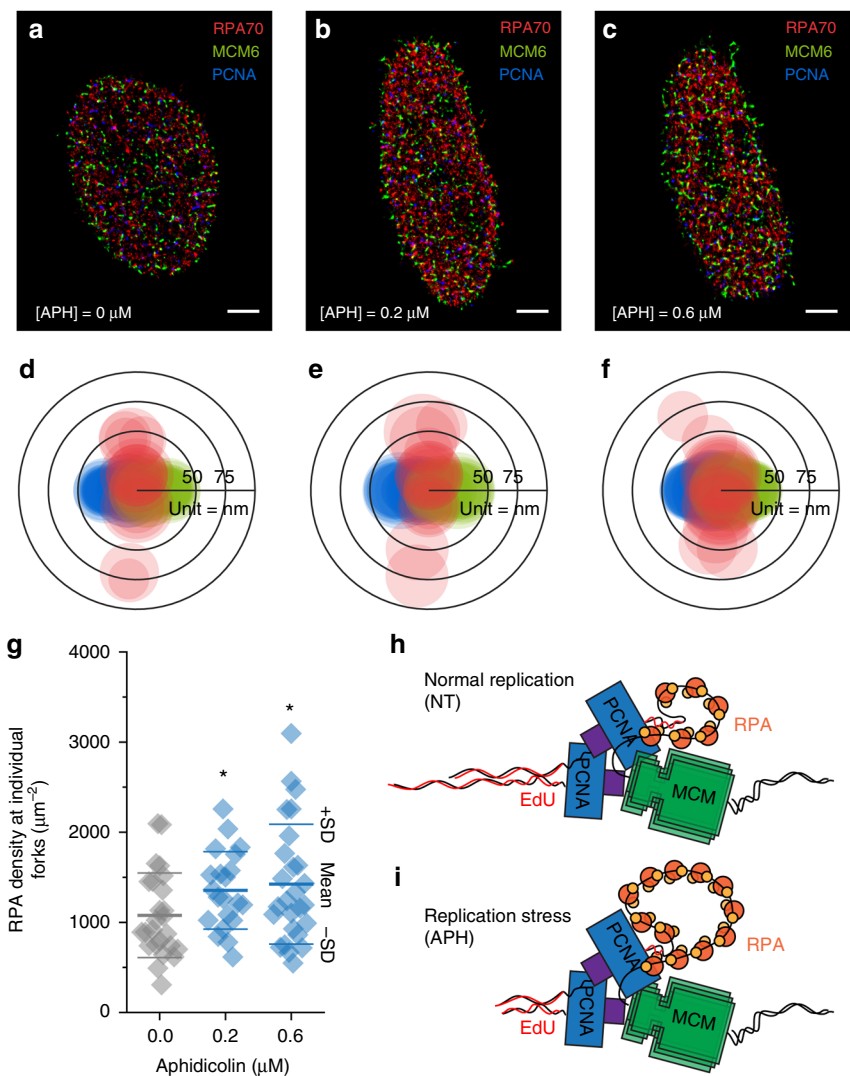

**Fig. 4** RPA local density at individual replication forks estimated by the TC function. **a–c** Representative U2OS nucleus labeled with RPA70 (R), PCNA (B), and MCM (G) were treated with 0, 0.2, and 0.6 µM Aphidicolin (APH), respectively. Scale bar = 2500 nm. **d–f** Configurations resolved from $N = 80$, 95, and 101 nuclei treated with 0, 0.2, and 0.6 µM APH, respectively (from 3 experimental replicates). Filtered by the criteria Dist_BG (distance between PCNA and MCM) $\leq$ 100 nm, 25, 21, and 26 configurations from [APH] = 0, 0.2, and 0.6 µM, respectively, were overlaid by aligning the BG edge onto the same horizontal, so that the relative distribution of PCNA (B) to RPA70 (R) and MCM (G) is clearly seen to form a fork-like pattern where RPA were mostly found in between the PCNA and MCM. The circle size at each vertex represents the local density of RPA at each individual fork as calculated by multiplying the conditional probability of finding RPA with each given pairwise PCNA-MCM and the overall average density of RPA. **g** Statistics of the average local density of RPA at each individual fork (as displayed as the size of the circles in **d–f**). Each plot displays such average local density from one nucleus. $p = 0.043$ and 0.036 for unpaired t-test between [APH] = 0 vs. [APH] = 0.2 µM and [APH] = 0 vs [APH] = 0.6 µM, respectively (*, unpaired t test, $0.01 \leq p < 0.05$). **h**, **i** A molecular model of RPA accumulating in between PCNA and MCM in non-treated control (Normal Replication, NT) cells (**h**) and APH stressed (Replication Stress, APH) cells (**i**). Source data for Fig. 4d-4j are provided as a Source Data file

$P$ (RPA|PM). Figure 4d–f are the overlaid PCNA-RPA-MCM configurations resolved from more than 80 nuclei at each APH concentration. These configurations clearly indicate a fork-like pattern where RPA-ssDNA were mostly found in between PCNA and MCM. Moreover, instead of representing the Triple-Correlation amplitude as in Fig. 3j, the circle size in Fig. 4d–f represents the RPA local density $\rho$ (RPA|PM), from which, a slight increase of RPA at each replication fork upon APH treatment were quantified (Fig. 4g, h), indicating the capability of the Triple-Correlation function in measuring the local molecular density at specific molecular configurations.

In summary, we have demonstrated a fast and robust algorithm for Triple-Correlation analysis of high-content multi-color SMLM images, providing a platform for high-throughput Triple-Correlation analyses of dense images. Alongside with further developments of multiplexed super-resolution imaging techniques, the dTC paves a way for more vigorous understanding of functional molecular architectures inside cells, especially for in-depth studies of biological metabolisms in their dense circumstances. We note that the theoretical accuracy of the TC function, as well as other image analyses methods are limited by the accuracy of the original SMLM data (Supplementary Figure 5).

## Methods

**Simulation**. The simulations in this work were performed as following (unless specifically stated otherwise): We randomly positioned and orientated the designed triangular configurations onto the canvas, and assigned the vertexes as the

positions of the "molecules". Around each of these "molecules", we simulated the SMLM data by generating multiple localization coordinates that subject to a 2D-Gaussian distribution centering at each of the "molecules" and broadening with the experimental localization precision as the standard deviation ($\sigma$) of the Gaussian profile.

**Sample preparation**. U2OS cells (ATCC HTB-96) were passaged onto glass coverslips and grown in DMEM (ThermoFisher 11965092) with 10% FBS (Gemini Bio 100-106) and 100 U/mL Penicillin-Streptomycin (ThermoFisher 15140) for 24–48 h until established. Cells were then synchronized to S-phase, via 72 h Serum withdrawal followed by 17 h incubation in full media. A concentration of 20 μM Hydroxyurea and EdU was introduced 2 h and 15 min, respectively, prior to the end of the 17 h incubation for experiments in Fig. 3 and Supplementary Figure 3; Aphidicolin was introduced 1 h prior to the end of the 17 h incubation for experiments in Fig. 4 and Supplementary Figure 4. U2OS cells were then permeabilized with 0.5% Triton in CSK buffer (10 mM Hepes, 300 mM Sucrose, 100 mM NaCl, and 3 mM MgCl$_2$, pH = 7.4) for 10 min, and fixed with paraformaldehyde (4%) for 30 min. The cells were then rinsed with PBS and incubated in blocking buffer (2% glycine, 2% BSA, 0.2% geltin, and 50 mM NH$_4$Cl in PBS) overnight at 4C. EdU was tagged with Alexa Fluor 647 picolyl azide through click reaction kit (ThermoFisher, C10640). RPA was stained by either Rabbit anti-RPA antibody (Abcam, ab79398) for 1 h at 1:1000 dilution at room temperature, followed by Alexa Fluor 750-conjugated anti-Rabbit antibody (ThermoFisher, A-21039) for 0.5 h at 1:10,000 dilution at room temperature (for experiments in Supplementary Figure 3), or Alexa Fluor 647 conjugated anti Rabbit antibody (Abcam, ab199240) for 1 h at 1:1000 dilution at room temperature (for experiments in Fig. 4 and Supplementary Figure 4). PCNA was immunostained by Alexa 488 conjugated anti-PCNA antibody (Abcam, ab201672) and MCM was immunostained by Alexa 568 conjugated anti-MCM antibody (Abcam, ab211916). Both antibodies were incubated for 1 h at 1:1000 dilution at room temperature. The fixed U2OS cells were then mounted onto microscope glass for single-molecule localization imaging in freshly mixed imaging buffer (1 mg/mL glucose oxidase, 0.02 mg/mL catalase, 10% glucose, and 100 mM cycteanube (MEA)).

**Optical setup and image acquisition**. The single-molecule localization imaging was performed on a customized Leica DMI 300 inverse microscope. A 750 nm laser (UltraLaser, MDL-III-750-500), 639 nm laser (UltraLaser, MRL-FN-639-800), 561 nm laser (UltraLaser, MGL-FN-561-200), and 488 nm Laser (OBIS) were aligned and reflected into an HCX PL APO 63X NA = 1.47 OIL CORR TIRF Objective (Zeiss) by a penta-edged dichroic beam splitter (FF408/504/581/667/762-Di01-22 × 29). The 488, 561, 639, and 750 laser lines were adjusted to ~0.8, 1.0, 1.5, and 0.4 kW/cm$^2$. A 405 nm Laser line (MDL-III-405-150, CNI) was also equipped to reactivate Alexa Fluor 647 fluorophores. The cell samples were sequentially illuminated, and their emitted fluorescence was also sequentially collected with single-band fluorescence filter switched in a filter wheel accordingly. In brief, the emitted fluorescence was collected by the same objective and further magnified by a 2X lens tube (Diagnostic Instruments). The fluorescence was then filtered by a single-band filter (Semrock FF01-531/40, FF01-607/36, and FF01-676/37 for Alexa Fluor 488, Alexa Fluor 568, and Alexa Fluor 647, respectively) and a chromatic aberration correction lens (AC254-300-A, Thorlabs), and collected by a sCMOS camera (Photometrics, Prime95B) at 33 Hz. 2000 frames were recorded for each color in each image stack. In particular, considering the patterned sCMOS camera, the readout noise of each pixel camera was pre-calibrated, and characterized by a Gaussian distribution. The expectation, variation, and the analog-to-digital conversion factor of such calibrations of each pixel was used in single-molecule localization as described in the Single-Molecule Localization section.

**Alignment of images from different colors**. Aligning images from different colors was performed by separately mapping blue (488), green (568), and dark red (750) onto the red (639) channel, using a 2nd polynomial mapping algorithm. In brief, broad-spectrum fluorescent beads (Diameter ~ 100 nm, TetraSpec, Thermofisher, note that the 750-channel mapping was accomplished by illuminating such beads using the 561 nm laser to collect sufficient signal-to-noise ratio of the bead images) were imaged on all the four-color channels. The mass centers of the same bead were recorded as vectors $\{x_i^{CHX}, y_i^{CHX}\}$, where $i$ denotes the $i$-th bead and CHX denotes the $X$-th channel, and submitted for 2nd polynomial optimization of the transform coefficient $\{K_j^{(x)}\}$ and $\{K_j^{(y)}\}$.

$$x_i^{CHR} = \sum_{j=0}^{8} K_j^{(x)} (x_i^{CHX})^l (y_i^{CHX})^m$$

$$y_i^{CHR} = \sum_{j=0}^{8} K_j^{(y)} (x_i^{CHX})^l (y_i^{CHX})^m,$$

where $l = \lfloor j/3 \rfloor$ is the maximum integer smaller than $j/3$ and $m = j - 3\lfloor j/3 \rfloor$ is the modulo of $j/3$; CHX denotes the channels other than the Red (reference) channel.

The optimized coefficient of the polynomial function was then applied to align the Blue, Green, and Dark Red real sample images to the Red channel. We note

that higher-order polynomial regression might result in better optimization, depending on the optical alignment and chromatic aberration of the experimental microscope setup. Higher than 2nd order regression in this study could cause overfitting. We also note that this polynomial regression sufficiently reduced the chromatic aberration in our measurements (Supplementary Figure 6).

**Single-molecule localization**. Each frame from an image stack was first box-filtered with the box size of 4 times of the FWHM of a 2D Gaussian PSF. We note that each pixel was weighted by the inverse of its variation during such box-filtering. The low-pass filtered image was then extracted from the raw image, followed by recognition of local maximums. The local maximums from all the frames of the image stack were then submitted for 2D-Gaussian single-PSF fitting.

The 2D-Gaussian single-PSF fitting were performed in GPU (Nvidia GTX 1060, CUDA 8.0) using the Maximum Likelihood Estimation (MLE) algorithm. In brief, the likelihood function at each pixel was built by convolving the Poisson distribution of the shot noise governed by the photons emitted from fluorophores nearby, and the gaussian distribution of the readout noise that characterized by the expectation, variation, and the analog-to-digital conversion factor that pre-calibrated as mentioned above. The fitting accuracy was estimated by Cramér-Rao lower bound (CRLB).

**Code availability**. Codes for the dTC and dPC algorithms, as well as a testing demo (with simulation codes) are available at https://github.com/yiny02/direct-Triple-Correlation-Algorithm. The code is for Research and Educational Purposes for Non-Profit Academic and/or Research Institutions.

**Reporting Summary**. Further information on experimental design is available in the Nature Research Reporting Summary linked to this article.

## Data availability

A reporting summary for this Article is available as a Supplementary Information file. The major source data underlying Figs. 2c, 3j, and 4d-g and Supplementary Figs 3j, 4d, and 6 are provided as a Source Data file. Other simulated and experimental data is available from the authors upon requests.

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

## Acknowledgements

We thank members of the Rothenberg laboratory for critically reading and commenting on the manuscript. Research in the Rothenberg lab is supported by funds from the NIH R01 GM108119, American Cancer Society (ACS: 130304-RSG-16-241-01-DMC), the V Foundation for Cancer Research (D2018-020), and Fondation Leducq (17CVD02).

## Author contributions

Y.Y. and E.R. conceived the project and designed algorithm. Y.Y. performed the simulations. Y.Y. and W.T.C.L performed the experiments. Y.Y. and W.T.C.L analyzed the simulated and experimental data. Y.Y. and E.R. wrote the manuscript.

## Additional information

**Competing interests:** The authors declare no competing interests.

