## [Peer Review File · Nature Communications]

Reviewers' comments:

Reviewer #1, expert in SMLM localisation algorithms (Remarks to the Author):

This manuscript describes well-executed work that significantly improves the speed of a useful analysis technique in superresolution microscopy. The key insight is that while direct point-by-point calculation of a triple correlation function is asymptotically slower than Fourier transform techniques (at high densities and for uniformly-distributed localizations the time scales as the third power of area, or sixth power of the linear size N of the imaging window, while Fourier transform techniques scale as N^4), if there's at least some degree of sparseness then the point-by-point calculation can be faster. This follows from the fact that for some range of N values, $a \cdot N^6 < b \cdot N^4$, where a and b are positive constants. (I'm leaving out the logarithms because they don't grow rapidly.) This is an important insight, and the authors demonstrate that this insight has meaningful consequences for improving the speed of image analysis. They also demonstrate that their algorithm can distinguish between dimers and trimers (supplementary figure 2). The work is competently executed, and worthy of publication, after one concern has been addressed.

My one methodological concern is that when the authors work with simulated data (results in Figure 2), it is not clear whether they included noise in the simulated coordinates. In a real superresolution experiment, the localizations do not correspond exactly to molecular positions, but rather to molecular positions with noise added. If the localization precision (root mean square noise) is comparable to the spacing of the molecules in the trimers then the ability to identify trimers and their geometry will depend on the sample size and just how large the noise is. Some analysis of this issue should be included, so that the prospective user can understand when this tool will be useful.

Points for clarification:

- 1) The authors show time comparisons but barely discuss WHY this algorithm is faster than the Fourier transform approach. For those of us who have mostly worked with pair correlations, the lesson to always compute correlations with Fourier transforms is a deeply-ingrained one, and it would be worthwhile to add a paragraph or two in the supplemental information, so that the reader can understand why this algorithm works so well, and why the degree of improvement decreases as the density of localizations increases.
- 2) Very little is said about the GPU implementation. Why is the time on a GPU of the same order of magnitude as the time on a CPU? I think I know why, but the reader would benefit from hearing why this task is not easily decomposed and parallelized.
- 3) Supplementary note 1, equation 1: The notation $\rho(r_{12}|r(\text{CH1})=0)$ seems unnecessary. Isn't it implicit that the sum is over read localizations, implying vectors drawn from red localizations, so that in each case the red localization is taken as the origin?
- 4) The text accompanying supplementary figure 2 starts off a bit confusing. I get the distinction between pair correlations and triple correlations, but I don't know why the authors say that the extraction of pair correlations is "hindered in the definition of triple correlation." The rest of the text makes an important point, but the first line is confusing.
- 5) Also in the text accompanying supplementary figure 2, what is g_{delta} ? It was confusing.

Minor points to correct:

I did not do thorough proofreading, but a few sentences jumped out at me as needing correction:

- 1) Main text, page 2, line 52: The sentence that begins with "Current method" would sound better if revised to "_Currently, the most common method_ for computing the Triple Correlation function is based on the bispectrum convolution theorem, which requires the detected localizations to be rendered from coordinates into binary pixels_,_ and performs _a_ 4D discrete Fourier Transform (FT)." (Suggested changes are between _ _ symbols.)
- 2) Main text, page 5, sentence starting on line 138: This sentence should probably be broken into two, rather than kept as one sentence with a semicolon.

3) Supplementary figure 1: The last two sentences should be a single sentence, with a comma before "where $b_{i+\theta}$ stands for..."

Reviewer #2, expert in image analysis algorithms (Remarks to the Author):

In this manuscript Yin et al. present a molecular characterization algorithm for multiplexed data that is orders of magnitude faster than previous methods.

The addressed topic is important and of an interest of the community. The language is nice to read, and easy to follow the paper. The code is well structured and documented. The mathematical derivations seem correct to me. The supplementary documents gave enough details to understand the work. Experiments are well designed, described and performed. I do not think the novelty of the proposed method is large.

I think the paper is already in a proper shape and I only had a few weaknesses to point out:

- Abstract does not tell me why to read this paper, I think at minimum it should have a sentence about the method and idea what the Authors propose
- Introduction gives only a little about the proposed method
- Methods section would benefit from describing the method better in details
- Alignment in the supplementary: It is not clear what type of registration was performed? I guess linear transformation. Also would be interesting to know if there is a chromatic aberration towards the edges of the image that could lead to misinterpretations?

Point-by-Point response to referees

Ultrafast Data Mining of Molecular Assemblies in Multiplexed High-Density Super-Resolution Images

Yandong Yin*, Wei Ting Chelsea Lee, and Eli Rothenberg*

Point-by-point reply to the reviewers' comments (authors reply is marked in blue):

Reviewer #1, expert in SMLM localization algorithms (Remarks to the Author):

This manuscript describes well-executed work that significantly improves the speed of a useful analysis technique in superresolution microscopy. The key insight is that while direct point-by-point calculation of a triple correlation function is asymptotically slower than Fourier transform techniques (at high densities and for uniformly-distributed localizations the time scales as the third power of area, or sixth power of the linear size N of the imaging window, while Fourier transform techniques scale as N^4), if there's at least some degree of sparseness then the point-by-point calculation can be faster. This follows from the fact that for some range of N values, $a \cdot N^6 < b \cdot N^4$, where a and b are positive constants. (I'm leaving out the logarithms because they don't grow rapidly.) This is an important insight, and the authors demonstrate that this insight has meaningful consequences for improving the speed of image analysis. They also demonstrate that their algorithm can distinguish between dimers and trimers (supplementary figure 2). The work is competently executed, and worthy of publication, after one concern has been addressed.

My one methodological concern is that when the authors work with simulated data (results in Figure 2), it is not clear whether they included noise in the simulated coordinates. In a real superresolution experiment, the localizations do not correspond exactly to molecular positions, but rather to molecular positions with noise added. If the localization precision (root mean square noise) is comparable to the spacing of the molecules in the trimers then the ability to identify trimers and their geometry will depend on the sample size and just how large the noise is. Some analysis of this issue should be included, so that the prospective user can understand when this tool will be useful.

- The reviewer has raised an important point regarding the impact of the SMLM noise on image analysis. Indeed, in the simulated data in Figure 2, we first established the positions of the molecules on the canvas, and then around each of these 'molecules', we simulated its 'experimental' localization coordinates by generating multiple random localizations that governed by a 2D Gaussian distribution centering at the 'molecule' and broadening with a given localization precision as its sigma. Although this simulation included the 'experimental' noise, we recognize the importance of further elaborating on how the localization precision (SMLM noise) affects the Triple-Correlation image analyses. To address this concern, we provide detailed analyses in **Supplementary Figure 5** in the revised manuscript, discussing the outcome of the Triple-Correlation analyses when the localization uncertainty is larger, comparable, and smaller than the spacing of trimer complexes.
- We also included a detailed description of the simulations in the revised **Supplementary Methods section**.

Points for clarification:

1) The authors show time comparisons but barely discuss WHY this algorithm is faster than the Fourier transform approach. For those of us who have mostly worked with pair correlations, the lesson to always compute correlations with Fourier transforms is a deeply-ingrained one, and it would be worthwhile to add a paragraph or two in the supplemental information, so that the reader can understand why this algorithm works so well, and why the degree of improvement decreases as the density of localizations increases.

- To address this point, we have included a detailed explanation on why this algorithm is faster. This is provided in the first paragraph of the **Supplementary Technical Discussion section**, and also as an additional Supplementary Note (**Supplementary Note 2**) to adapt this algorithm to computing the pair-correlation function.

2) Very little is said about the GPU implementation. Why is the time on a GPU of the same order of magnitude as the time on a CPU? I think I know why, but the reader would benefit from hearing why this task is not easily decomposed and parallelized.

- To address this point, in the second paragraph of the **Supplementary Technical Discussion section**, we provide a description as to why GPU is not able to further accelerate this algorithm.

3) Supplementary note 1, equation 1: The notation $\rho(r_{12}|r_{CH1}=0)$ seems unnecessary. Isn't it implicit that the sum is over read localizations, implying vectors drawn from red localizations, so that in each case the red localization is taken as the origin?

- We thank the reviewer for noting this. We corrected and reformulated the Equation in the revised **Supplementary Note 1**.

4) The text accompanying supplementary figure 2 starts off a bit confusing. I get the distinction between pair correlations and triple correlations, but I don't know why the authors say that the extraction of pair correlations is "hindered in the definition of triple correlation." The rest of the text makes an important point, but the first line is confusing.

5) Also in the text accompanying supplementary figure 2, what is g_{δ} ? It was confusing.

- The reviewer raised an important point with respect to the definition of the correlation functions.

The pair-correlation function is defined as either $g^{\text{def1}}(\vec{r}) = \langle \rho_{\text{CHX}}(\vec{R}) \rho_{\text{CHY}}(\vec{R} + \vec{r}) \rangle_{\vec{R}} / (\langle \rho_{\text{CHX}} \rangle_{\vec{R}} \langle \rho_{\text{CHY}} \rangle_{\vec{R}})$ or $g^{\text{def2}}(\vec{r}) = \langle \delta \rho_{\text{CHX}}(\vec{R}) \delta \rho_{\text{CHY}}(\vec{R} + \vec{r}) \rangle_{\vec{R}} / (\langle \rho_{\text{CHX}} \rangle_{\vec{R}} \langle \rho_{\text{CHY}} \rangle_{\vec{R}})$, where $\delta \rho_{\text{CHX}}(\vec{R}) = \rho_{\text{CHX}}(\vec{R}) - \langle \rho_{\text{CHX}} \rangle_{\vec{R}}$, and channel X (CHX) and channel Y (CHY) refers to the same or two different signal channels for auto- or cross-correlation, respectively. Since $g^{\text{def1}}(\vec{r}) \equiv g^{\text{def2}}(\vec{r}) + 1$, these two different definitions are both widely accepted and applied.

However, these definitions are quite different in Triple-Correlation. As shown in the derivation in the revised **Supplementary Note 1** and simulation in **Supplementary Figure 2**, $g^{\text{def1}}(\vec{r})$ results in a mixture of trimers and dimers whereas $g^{\text{def2}}(\vec{r})$ only provides trimer information.

We realized that the phrasing 'hindered in the definition of the triple correlation' was confusing, and restated and clarified this part in the revised **Supplementary Note 1**. We moved the derivation from the legend of the Supplementary Figure 2 to the **Supplementary Note 1**, and clarified that the g_{δ} denotes the g^{def2} accordingly.

Minor points to correct: I did not do thorough proofreading, but a few sentences jumped out at me as needing correction:

- 1) Main text, page 2, line 52: The sentence that begins with “Current method” would sound better if revised to “_ Currently, the most common method_ for computing the Triple Correlation function is based on the bispectrum convolution theorem, which requires the detected localizations to be rendered from coordinates into binary pixels,_ and performs _a_ 4D discrete Fourier Transform (FT).” (Suggested changes are between __ symbols.)
- 2) Main text, page 5, sentence starting on line 138: This sentence should probably be broken into two, rather than kept as one sentence with a semicolon.
- 3) Supplementary figure 1: The last two sentences should be a single sentence, with a comma before “where $b_{i+\theta}$ stands for...”

- We thank the reviewer for these important suggestions, and have incorporated changes accordingly, as well as additional proofreading.

Reviewer #2, expert in image analysis algorithms (Remarks to the Author):

In this manuscript Yin et al. present a molecular characterization algorithm for multiplexed data that is orders of magnitude faster than previous methods. The addressed topic is important and of an interest of the community. The language is nice to read, and easy to follow the paper. The code is well structured and documented. The mathematical derivations seem correct to me. The supplementary documents gave enough details to understand the work. Experiments are well designed, described and performed. I do not think the novelty of the proposed method is large.

I think the paper is already in a proper shape and I only had a few weaknesses to point out:

- Abstract does not tell me why to read this paper, I think at minimum it should have a sentence about the method and idea what the Authors propose.
- Introduction gives only a little about the proposed method
- Methods section would benefit from describing the method better in details

- We thank the reviewer for the helpful comments, and have addressed this in the revised manuscript, providing detailed descriptions and explanations in several sections (SI).

- Alignment in the supplementary: It is not clear what type of registration was performed? I guess linear transformation. Also would be interesting to know if there is a chromatic aberration towards the edges of the image that could lead to misinterpretations?

- To address this point, we provide additional information regarding channel alignment (via polynomial transformation) in the revised **Supplementary Methods** section. We added **Supplementary Figure 6** in the revised manuscript. The Supplementary Figure 6 includes statistical evaluations of the alignment accuracy, and the corrections of the chromatic aberration as a function of the distance to the center of the imaging area. As shown in **Supplementary Figure 6**, the alignment used in this manuscript have efficiently reduced the chromatic aberration.

REVIEWERS' COMMENTS:

Reviewer #1 (Remarks to the Author):

I am satisfied with the changes. My concerns have been addressed, and I have no further concerns. I recommend publication.

Reviewer #2 (Remarks to the Author):

Authors answered my comments, the Abstract and the Introduction has been extended and I think more appropriate. A new Supplementary has been added explaining the registration. I agree to publish this work. Thank you!

Ultrafast Data Mining of Molecular Assemblies in Multiplexed High-Density Super-Resolution Images

Yandong Yin*, Wei Ting Chelsea Lee, and Eli Rothenberg*

Point-by-point reply to the reviewers' comments (authors reply is marked in blue):

Reviewer #1, expert in SMLM localisation algorithms (Remarks to the Author):

I am satisfied with the changes. My concerns have been addressed, and I have no further concerns. I recommend publication.

Reviewer #2, expert in image analysis algorithms (Remarks to the Author):

Authors answered my comments, the Abstract and the Introduction has been extended and I think more appropriate. A new Supplementary has been added explaining the registration. I agree to publish this work.

We thank the reviewers for their helpful comments that have improved the quality of our manuscript.